materials science/nanotechnology

MXene, MXene nanocomposites, reflection loss, macroscopic design

**Author for correspondence:**
Daniel Q. Tan
e-mail: daniel.tan@gtiit.edu.cn

This article has been edited by the Royal Society of Chemistry, including the commissioning, peer review process and editorial aspects up to the point of acceptance.

# Enhancement of microwave absorption bandwidth of MXene nanocomposites through macroscopic design

Pritom J. Bora[1], T. R. Suresh Kumar[2,3] and Daniel Q. Tan[1]

[1]Department of Materials Science and Engineering, Technion Israel Institute of Technology and Guangdong Technion Israel Institute of Technology, Shantou, Guangdong Province, People's Republic of China 515063
[2]Department of Electrical and Communication Engineering, Indian Institute of Science, Bengaluru 560012, India
[3]Department of Electronics Engineering, Vellore Institute of Technology, Vellore 632014, Tamil Nadu, India

DQT, 0000-0002-2282-2000

MXene, the new family of two-dimensional materials having numerous nanoscale layers, is being considered as a novel microwave absorption material. However, MXene/functionalized MXene-loaded polymer nanocomposites exhibit narrow reflection loss (RL) bandwidth (RL less than or equal to −10 dB). In order to enhance the microwave absorption bandwidth of MXene hybrid-matrix materials, for the first time, macroscopic design approach is carried out for $TiO_2$-$Ti_3C_2T_x$ MXene and $Fe_3O_4$@$TiO_2$-$Ti_3C_2T_x$ MXene hybrids through simulation. The simulated results indicate that use of pyramidal meta structure of MXene can significantly tune the RL bandwidth. For optimized MXene hybrid-matrix materials pyramid pattern, the bandwidth enhances to 3–18 GHz. Experimental RL value well matched with the simulated RL. On the other hand, the optimized $Fe_3O_4$@$TiO_2$-$Ti_3C_2T_x$ hybrid exhibits two specific absorption bandwidths (minimum RL value - −47 dB). Compared with other two-dimensional nanocomposites such as graphene or $Fe_3O_4$-graphene, MXene hybrid-matrix materials show better microwave absorption bandwidth in macroscopic pattern.

## 1. Introduction

In the recent years, the cutting-edge telecommunication, healthcare systems, detective systems, military applications, etc., predominantly use microwave frequency [1–7]. However, due to

**Figure 1.** (a) Schematic of as-synthesized MXene and the structure of $Fe_3O_4@TiO_2-Ti_3C_2T_x$. (b,c) Surface morphology of $TiO_2-Ti_3C_2T_x$ MXene and (d,e) $Fe_3O_4@TiO_2-Ti_3C_2T_x$ MXene.

the presence of electromagnetic interference (EMI) or electromagnetic threat, the need of microwave absorbing materials is increasingly felt especially in the frequency range 2–18 GHz [2–4]. The traditional ferrites-based and carbon-based absorbers have many disadvantages such as low specific absorption, corrosion and narrow bandwidth [4–10]. Polymer-based EMI absorbers using different fillers such as conducting and magnetic nanoparticles are also not capable of offering high broadband microwave absorption as a single layer because of low impedance match and quarter wavelength resonance [10]. In search of novel ultra-light, broadband microwave absorption materials, great interest has recently been directed to the new family of two-dimensional materials and functionally reinforced composites [11]. MXene has been particularly noted by the scientific community in the last 8 years for its unique structure and electrical properties [12–14]. MXene's utilization for microwave absorption is increasing day by day; however, the understanding of its contribution is only related to its metal-like properties and surface-rich functional groups [1,11–18]. In addition, the graphene-like two-dimensional materials with a multi-layer feature, MXene has a high dielectric loss in a polymer matrix at low percolation limit (15 wt%) [12]. Further, lightweight MXene (density 0.029 $gcm^{-3}$) also has been proposed for microwave absorption in X-band [13]. Yet, the potential of MXene, its orientational structure and impacting roles in composites are to be explored with more computer simulation [11]. It should be noted that for better microwave absorption the combination of moderate conductivity and magnetic-dielectric system has benefits because of its impedance matching and high electromagnetic energy loss. Introduction of certain oxidation state of MXene will be more advantageous as it decreases the conductivity and increases the polarization defects within MXene. Therefore, it can be considered as an effective approach for enhanced microwave absorption property of MXene. The reported reflection loss (RL) of MXene-based polymer nanocomposites since its discovery (2011) is tabulated in the literature [12]. From the literature, it is clear that single-layer MXene composites possess excellent RL, but very narrow bandwidth (for practical application RL should be less than or equal to −10 dB). In fact, most of the reported microwave absorber exhibits very narrow bandwidth [19,20]. In order to enhance the RL bandwidth, Liu *et al.* [10] reported the macroscopic design, where as a case study, graphene and magnetic graphene were used. However, the macroscopic design of MXene nanocomposites for enhanced bandwidth has not been reported. In this work, we have designed a macroscopic structure of MXene nanocomposites based on impedance matching, simulated the RL and investigated the RL bandwidth advantage.

# 2. Results and discussion

The schematic of the synthesized $TiO_2-Ti_3C_2T_x$ MXene and $Fe_3O_4$-coated MXene ($Fe_3O_4@TiO_2-Ti_3C_2T_x$) is shown in figure 1a (synthesis details are explained in the Method). Figure 1b,c, shows the surface morphology of the synthesized $TiO_2-Ti_3C_2T_x$ MXene. The lateral dimension of $TiO_2-Ti_3C_2T_x$ MXene was obtained to be 1–9 µm with minimal delamination. The surface morphology of hydrothermally ultra-small $Fe_3O_4$ nanoparticle-coated $TiO_2-Ti_3C_2T_x$ MXene ($Fe_3O_4@TiO_2-Ti_3C_2T_x$) is shown in figure 1c,d. The energy-dispersive X-ray (EDX) confirms the presence of Fe (electronic supplementary material, figure S1). Figure 1c,d also suggests that the $Fe_3O_4$ nanoparticles bridge the interlayers of

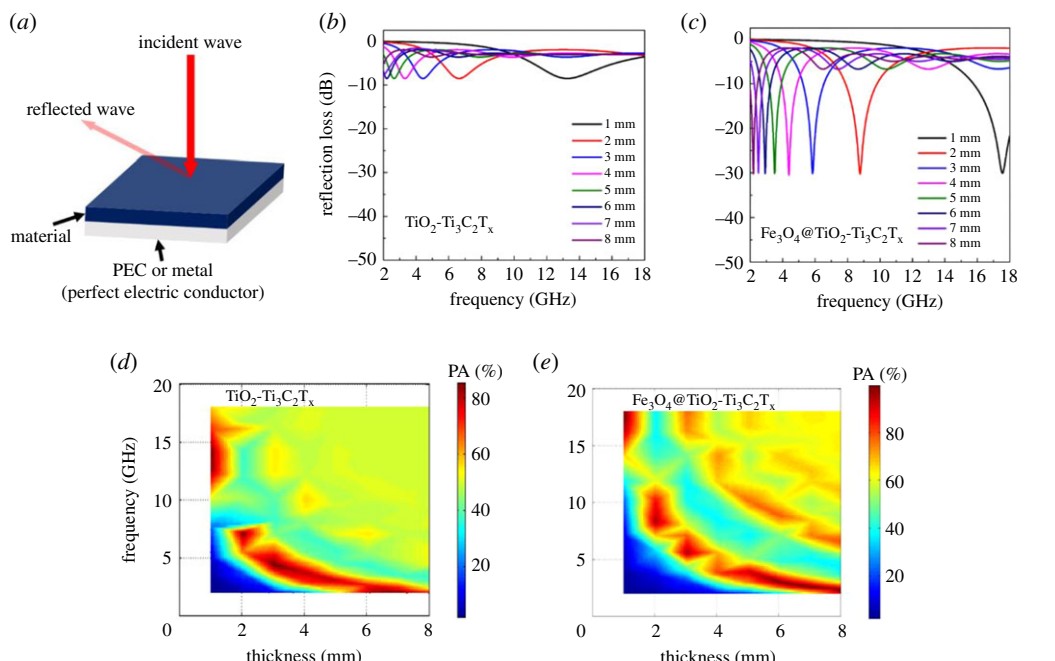

**Figure 2.** (a) Schematic of single-layer MXene hybrid-matrix materials. Simulated reflection loss (dB) of (b) $TiO_2$-$Ti_3C_2T_x$ and (c) magnetic $Fe_3O_4$@$TiO_2$-$Ti_3C_2T_x$ MXene-loaded paraffin in the frequency range of 2–18 GHz. Simulated EM power accepted (PA %) by the (d) paraffin-$TiO_2$-$Ti_3C_2T_x$ and (e) paraffin-$Fe_3O_4$@$TiO_2$-$Ti_3C_2T_x$ MXene hybrid in the frequency range of 2–18 GHz.

$TiO_2$-$Ti_3C_2T_x$. The $Fe_3O_4$ nanoparticle size was recorded using ImageJ software from SEM image, and a histogram is shown in the electronic supplementary material, figure S2. The average particle size of the $Fe_3O_4$ nanoparticles was found to be approximately 8 nm. The interaction of $Fe_3O_4$ nanoparticles with MXene is explained in [17]. The exposed hydroxyl groups offer the possibility of binding with MXenes along with blending metal–oxygen stretching modes such as Fe–O and Ti–O [17]. The magnetic properties of $Fe_3O_4$@$TiO_2$-$Ti_3C_2T_x$ MXene nanocomposite were explained elsewhere [17,18]. The RL (dB) of a perfect electric conductor (PEC)-backed material (figure 2a) can be expressed as [12–19]

$$\text{reflection loss (RL)} = 20\log\left|\frac{Z_{in} - Z_0}{Z_{in} + Z_0}\right|(\text{dB}),\tag{2.1}$$

where $Z_{in}$ is the input impedance and can be written as

$$Z_{in} = Z_0\sqrt{\frac{\mu_r}{\varepsilon_r}}\tanh\left(j\frac{2\pi fd\sqrt{\mu_r\varepsilon_r}}{c}\right),\tag{2.2}$$

where $Z_0$ is the characteristic impedance of free space (=377 Ω), $f$ is the frequency, $d$ is the thickness of the absorbing material and $c$ is the velocity of light ($3\times10^8$ m s$^{-1}$). $\varepsilon_r$ and $\mu_r$ are the relative permittivity ($\varepsilon_r = \varepsilon' - i\varepsilon''$) and permeability ($\mu_r = \mu' - i\mu''$), respectively. The real permittivity and imaginary permittivity values of paraffin-$TiO_2$-$Ti_3C_2T_x$ and paraffin-$Fe_3O_4$@$TiO_2$-$Ti_3C_2T_x$ hybrids are shown in electronic supplementary material, figure S3a and Figure S3b, respectively. The real permeability and imaginary permeability values of paraffin-$TiO_2$-$Ti_3C_2T_x$ and paraffin-$Fe_3O_4$@$TiO_2$-$Ti_3C_2T_x$ hybrids are shown in electronic supplementary material, figure S3c and Figure S3d, respectively.

The RL of a single-layer paraffin-$TiO_2$-$Ti_3C_2T_x$ and paraffin-$Fe_3O_4$@$TiO_2$-$Ti_3C_2T_x$ MXene nanocomposites was simulated (details in the Method) in the frequency range 2–18 GHz as shown in figure 2b,c, respectively. The simulated results indicate that the MXene nanocomposites were not effective for broadband absorption till the thickness was increased to 8 mm, whereas magnetic MXene ($Fe_3O_4$@$TiO_2$-$Ti_3C_2T_x$) hybrids had minimum RL value −30 dB. With increasing thickness, the RL value shifted to the low-frequency region; however, bandwidth remains narrow. This was due to the quarter wavelength resonance [10,21]. In order to enhance the RL bandwidth, the stored energy of the nanocomposites intrinsically should be high [18,22]. The power accepted (PA) by the MXene and magnetic MXene hybrids were also simulated [20,22]. Figure 2e,f shows the power accepted by the MXene and magnetic MXene hybrids, respectively. The electromagnetic power accepted by the

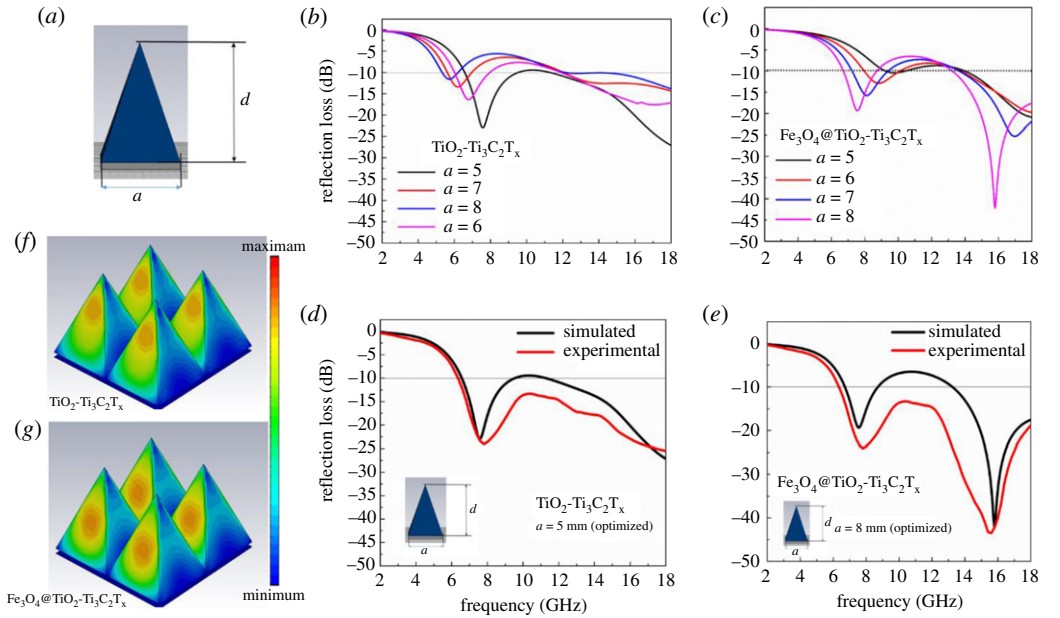

**Figure 3.** (a) Schematic of pyramid meta structure. (b,c) Simulated reflection loss (dB) of $TiO_2$-$Ti_3C_2T_x$ and $Fe_3O_4@TiO_2$-$Ti_3C_2T_x$ MXene hybrids for different 'a' value in the frequency range of 2–18 GHz respectively. (d,e) Comparison of simulation and experimental reflection loss (dB) of $TiO_2$-$Ti_3C_2T_x$ and $Fe_3O_4@TiO_2$-$Ti_3C_2T_x$ MXene hybrids, respectively, for optimized a value. (f–g) Power loss distribution of the $TiO_2$-$Ti_3C_2T_x$ and $Fe_3O_4@TiO_2$-$Ti_3C_2T_x$ MXene hybrids, respectively.

$Fe_3O_4@TiO_2$-$Ti_3C_2T_x$ MXene hybrids is intrinsically predominant as compared with the non-magnetic MXene ($TiO_2$-$Ti_3C_2T_x$) hybrid. This indicates that a single-layer magnetic MXene hybrid-matrix exhibits better PA (%) capability. In other words, it indicates the stored EM energy of $Fe_3O_4@TiO_2$-$Ti_3C_2T_x$ MXene hybrids is intrinsically predominant, especially for high thickness (approx. 8 mm).

It is always a challenge to get a wide bandwidth from a single-layer traditional uniform bulky composite [4–13]. Traditionally, the classic pyramid structure is well known for broadband microwave absorption for intrinsic wave impedance. In order to enhance the RL bandwidth of MXene nanocomposites, we designed the classic pyramid unit cell at the millimetre scale and a standard microwave simulation was carried out for the designed artificial array. The designed parameters are shown in figure 3a, the variable parameter 'a' corresponds to the width of the bottom and a height (d) was fixed at 8 mm. The value of 'a' was swept from 1 to 10 mm, and the optimized values were found to be 5, 6, 7 and 8. As shown in figure 3b, the RL and RL bandwidth of $TiO_2$-$Ti_3C_2T_x$ MXene hybrid was enhanced significantly, viz. 6–18 GHz for a = 5 mm. The simulated RL value of designed $Fe_3O_4@TiO_2$-$Ti_3C_2T_x$ MXene hybrid is shown in figure 3c for various 'a' values. The minimum RL value of the $Fe_3O_4@TiO_2$-$Ti_3C_2T_x$ MXene was −46 dB with two specific bandwidths, viz. low frequency (6.2–8.3 GHz) and high frequency (12.5–18 GHz). In order to validate the simulated results, the experimentally recorded data (measurement details are explained in electronic supplementary material) is compared in figure 3d,e for optimized 'a' values, viz. a = 5 mm for $TiO_2$-$Ti_3C_2T_x$ MXene hybrid and a = 7 and a = 8 mm for $Fe_3O_4@TiO_2$-$Ti_3C_2T_x$ MXene hybrid. The experimentally recorded RL data are well matched with the simulated results. In fact, experimental RL are more promising for all the MXene hybrids (figure 3e). The observed variation difference between simulated RL and experimentally recorded RL are due to many factors, viz. raw data fitting error, structure fabrication error. The experimental data confirm that the −10 dB RL bandwidth is guaranteed by both $TiO_2$-$Ti_3C_2T_x$ MXene and $Fe_3O_4@TiO_2$-$Ti_3C_2T_x$ MXene hybrid in the frequency range 6–18 GHz. However, minimum RL (approx. −43 dB) is predominant for optimized $Fe_3O_4@TiO_2$-$Ti_3C_2T_x$ MXene hybrid meta structure. The three-dimensional power loss distribution of the designed artificial periodic pattern of the MXene nanocomposites is shown in figure 3f and g, respectively, for non-magnetic and magnetic MXene hybrid. Clearly, it indicated that the energy loss in the edge region of the pyramid unit cell was predominant.

The alternate artificial multi-layer pattern design of the MXene hybrids was also carried out to investigate the effect of RL bandwidth. As shown in figure 4a, the thickness (8 mm) was divided into multi-layers (8 layers) having thicknesses of 1 mm each. The magnified parameter 'm' was assigned to

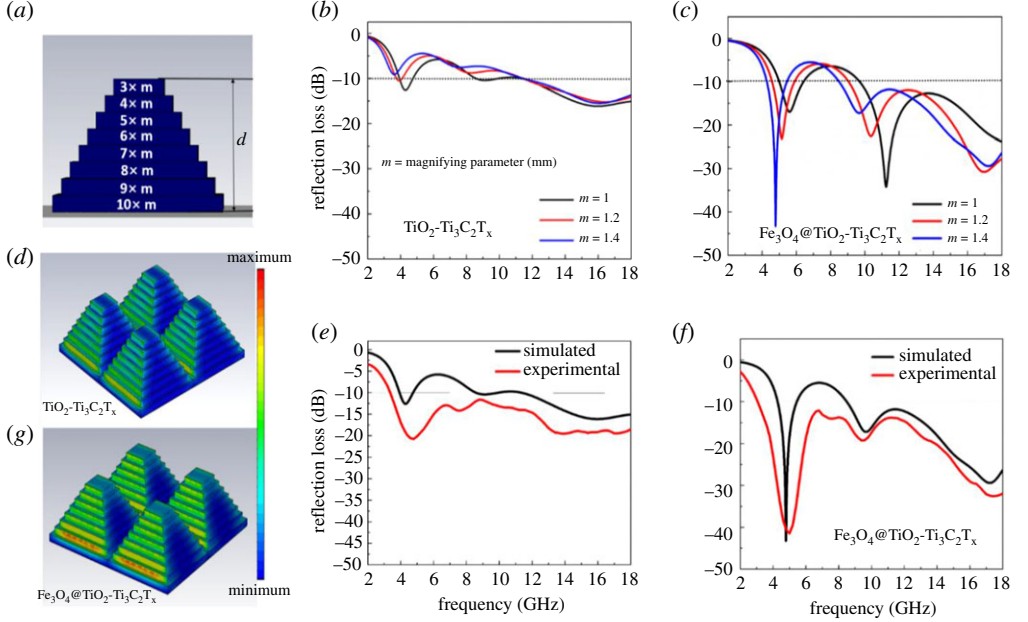

**Figure 4.** (a) Schematic of multi-layered pyramid meta structure. (b,c) Simulated reflection loss (dB) of $TiO_2$-$Ti_3C_2T_x$ and $Fe_3O_4@TiO_2$-$Ti_3C_2T_x$ MXene hybrid for different '$m$' values in the frequency range of 2–18 GHz, respectively. (d) Power loss distribution of the $TiO_2$-$Ti_3C_2T_x$ MXene hybrid. (e,f) Comparison of simulation and experimental reflection loss (dB) of $TiO_2$-$Ti_3C_2T_x$ and $Fe_3O_4@TiO_2$-$Ti_3C_2T_x$ MXene hybrids, respectively, for optimized $m$ value ($m = 1.4$ mm). (g) Power loss distribution of the $Fe_3O_4@TiO_2$-$Ti_3C_2T_x$ MXene hybrid.

tuning the unit cell size in the artificial array. The simulated RL values of non-magnetic and magnetic MXene hybrids for different '$m$' values are shown in figure 4b and c, respectively. For optimized $m =$ 1.4 mm, interestingly, unlike graphene-based material, as shown in figure 4c, the RL value and the bandwidth of magnetic MXene hybrids were more predominant in the low frequency (below 6 GHz) and high frequency based on intrinsic wave impedance. Comparison of simulated RL results of two structures, viz. macroscopic pyramid and multi-layered pyramid with the same width (10 mm) of the bottom is shown in the electronic supplementary material, figure S4 for $Fe_3O_4@TiO_2$-$Ti_3C_2T_x$ MXene hybrids. Clearly, multi-layered pyramidal meta structure possesses better absorption bandwidth (electronic supplementary material, figure S4). The experimentally recorded RL of fabricated (electronic supplementary material) optimized multi-layered $TiO_2$-$Ti_3C_2T_x$ MXene hybrid ($m = 1$ mm) and $Fe_3O_4@TiO_2$-$Ti_3C_2T_x$ MXene hybrid ($m = 1.4$ mm) is shown in figure 4e and f, respectively. The variation of experimental results as compared with the simulated results is believed to be due to the error in raw data fitting and structure fabrication.

The experimental RL of both the optimized $TiO_2$-$Ti_3C_2T_x$ MXene hybrid and $Fe_3O_4@TiO_2$-$Ti_3C_2T_x$ MXene hybrid covers the −10 dB absorption bandwidth in the frequency range of 3–18 GHz. However, unlike $TiO_2$-$Ti_3C_2T_x$ MXene hybrid meta structure, minimum RL of $Fe_3O_4@TiO_2$-$Ti_3C_2T_x$ MXene-based hybrid is more predominant (minimum RL value approximately −47 dB). The three-dimensional power loss distribution of the designed artificial multi-layered periodic macroscopic pattern of the MXene hybrids is shown in figure 4d and g, respectively, for non-magnetic and magnetic MXene hybrids. As expected from previous design (figure 3e,f), the power loss is predominant at the edges of each layer of the multi-layered macroscopic pyramid, especially for $Fe_3O_4@TiO_2$-$Ti_3C_2T_x$ MXene hybrid (figure 4g). Thus, it leads to minimum RL value −47 dB in middle of C-band (4–6 GHz) and −30 dB for both X-band (8.2–12.4 GHz) and Ku-band (12.4–18 GHz), respectively. A comparison of the RL bandwidth of the optimized graphene-based and MXene-based hybrids with structural modifications are tabulated in the electronic supplementary material, tables S1 and S2 for pyramidal and multi-layered pyramidal macroscopic design, respectively. From, electronic supplementary material, tables S1 and S2, it is clear that MXene hybrids are more advantageous for broadband microwave absorption than graphene-based nanocomposites through macroscopic design.

In the three types of the designed MXene hybrid materials, viz. the traditional uniform single-layer bulk, pyramid unit cell array, and multi-layer pyramid unit cell array, each pattern was observed to possess specific microwave absorption performances. This is because each pattern has a unique

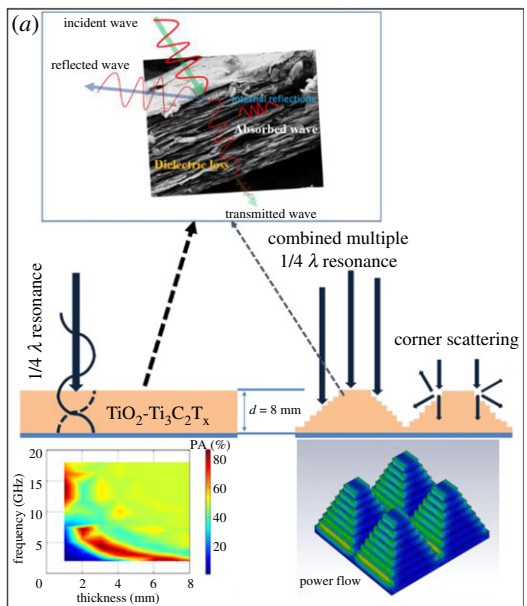
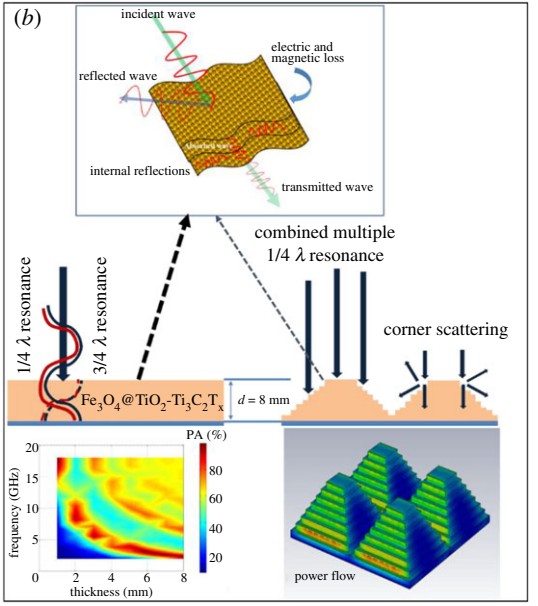

**Figure 5.** Scheme of microwave absorption mechanism in (*a*) non-magnetic (TiO$_2$-Ti$_3$C$_2$T$_x$) and (*b*) magnetic MXene (Fe$_3$O$_4$@TiO$_2$-Ti$_3$C$_2$T$_x$) hybrid-based array pattern.

absorption mechanism. The schematic of absorption mechanism is shown in figure 5. The uniform single-layer bulk microwave dielectric (paraffin-TiO$_2$-Ti$_3$C$_2$T$_x$) could generate quarter wavelength resonance depending on absorber thickness, and due to that for smaller thicknesses effective absorption takes place in high-frequency region (figure 5*a*). In the case of Fe$_3$O$_4$@TiO$_2$-Ti$_3$C$_2$T$_x$ MXene hybrid, 3/4 wavelength resonance also took place along with a quarter wavelength resonance due to the presence of magnetic component [4,12]. In addition, it also possesses a better impedance matching. In the case of pyramidal meta structure, the millimetre range pyramid interfaces played the key role, and the impedance matching at the top was maximized and gradually decreased. In the case of MXene hybrid-matrix, the charges in the interfaces (either heterogeneous or homogeneous interface) will generate aggregation and the rearrangement. It results in the polarization loss, which has a significant impact on dielectric loss [21]. Han *et al*. [23] described that the incorporation of TiO$_2$ and associated functional groups of MXene could generate dielectric dipole interactions at multiple interfaces and form capacitor-like structures. Therefore, it gave rise to an excellent stored electromagnetic energy throughout the pyramid interfaces by resulting in the enhancement of bandwidth especially in the high-frequency region. However, a real challenge was to achieve minimum RL value and bandwidth in the low-frequency region (below 6 GHz). As shown in figure 5*b*, in the case of multi-layer pyramidal structure, the synergistic effect of thicknesses, internal multiple reflections, corner scattering and different mode of resonances favoured the effective absorption of Fe$_3$O$_4$@TiO$_2$-Ti$_3$C$_2$T$_x$ MXene hybrid at low frequency as well as high frequency.

## 3. Conclusion

In summary, MXene as a superior conducting two-dimensional material was discovered to be advantageous in EMI absorption material design through a computer simulation. Leveraging the macroscopic pattern comprising a pyramidal meta structure design on the surface of polymer composites, MXenes enabled an excellent EMI absorption effectiveness and broadband performance. The tunability of the RL bandwidth became even better (6–18 GHz) when optimizing TiO$_2$-Ti$_3$C$_2$T$_x$ MXene hybrid pyramid pattern. Simulation results were verified with experimental data. Incorporating magnetic Fe$_3$O$_4$ nanoparticles, TiO$_2$-Ti$_3$C$_2$T$_x$ MXene hybrid exhibited a better microwave absorption bandwidth in macroscopic pattern compared with other two-dimensional nanocomposites such as graphene or Fe$_3$O$_4$-graphene, rendering a promising pathway for materials design exhibiting an effective electromagnetic interference shielding.

# 4. Methods

The standard synthesis method (as reported in [17,18]) was used to synthesize $TiO_2$-$Ti_3C_2T_x$ MXene. $Ti_3AlC_2$ powder (200 mesh, greater than or equal to 99% purity) was procured from Forsman Scientific Co., Ltd, China. In a typical synthesis, 10 g $Ti_3AlC_2$ powder was dispersed in 50 wt% HF solution with continuous stirring at $60 \pm 2°C$ for 48 h. After that, the suspension was centrifuged and washed several times with deionized water until pH reached approximately 6. Then, 5 wt% NaOH solution was added and sonicated for 1 h. Finally, it was dried at $60 \pm 2°C$ under vacuum. The $Fe_3O_4$ nanoparticles were coated over $TiO_2$-$Ti_3C_2T_x$ MXene by standard hydrothermal method [17]. In a typical synthesis, $FeCl_3.6H_2O$ and $NaHCO_3$ were dissolved in deionized (DI) water. After that, an aqueous solution of ascorbic acid with a molar ratio of $Fe^{3+}$ of $1:6$ was added into the above solution. The MXene suspension and $Fe^{3+}$ mixture was transferred into a Teflon-lined autoclave for a hydrothermal process at $150 \pm 5°C$ for 5 h. Finally, the powder was dried in a vacuum oven at $60 \pm 3°C$ for 24 h [17]. Details of the composite preparation and characterizations are given in the electronic supplementary material.

## 4.1. Computational method

From equation (2.1), it is clear that $\varepsilon_r$ and $\mu_r$ are the key factors for electromagnetic wave absorption, and a material having significant $\varepsilon_r$ and $\mu_r$ value is advantageous for impedance matching as well as better absorption. Moreover, the factor $\partial Z_{in}/\partial d$ is more sensitive for moderate permittivity and permeability [24]. Thus, it indicates that the reflection loss will be more sensitive with respect to thickness for moderate $\varepsilon_r$ and $\mu_r$ values. The optimized real and imaginary permittivity/permeability values of $TiO_2$-$Ti_3C_2T_x$ and coated $Fe_3O_4@TiO_2$-$Ti_3C_2T_x$ MXene-loaded paraffin nanocomposites were reported in the literature [17,18]. In this study, the same $\varepsilon_r$ and $\mu_r$ values as reported in [18] were used for electromagnetic simulation. Standard electrodynamic simulations for MXene nanocomposites were carried out using the commercial software Computer Simulation Technology (CST), Microwave Studio (2015). It is one of the most powerful electromagnetic computational tool and it solves Maxwell equations by resorting to the finite integration technique (FIT) in time domain and to a finite-element method (FEM) in the frequency domain [25]. In the designed structure, the unit cell with PEC substrate was constructed with same area ($10 \times 10$ mm) in the periodic array. The boundary conditions, viz. electric and magnetic, respectively, were applied at the X- and Y-direction, so that microwave propagates along the Z-axis. In order to validate the simulation results, optimized macroscopic pattern was fabricated for paraffin-MXene nanocomposites through solution processing and free space antenna measurements were done inside anechoic chamber (details are explained in the electronic supplementary material).

Data accessibility. This article does not contain any additional data.

Authors' contributions. P.J.B. and D.Q.T. designed the idea and entire work. T.R.S.K. performed simulations with P.J.B. and was also involved in the measurements. P.J.B. also did synthesis and fabrications, and prepared the manuscript. D.Q.T. has supervised the work and critically revised the manuscript.

Competing interests. We declare we have no competing interests.

Funding. This work was also supported by the Guangdong Basic and Applied Basic Research Foundation – 2019A1515012056.

Acknowledgement. Authors would like to thank Prof. Praveen C. Ramamurthy, Department of Materials Engineering, Indian Institute of Science (IISc) and Prof. K. J. Vinoy, Department of ECE, IISc, India for providing lab facility for characterizations and measurements.

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
