## [Reviewer comments · Royal Society Open Science]

Review History

RSOS-200456.R0 (Original submission)

Review form: Reviewer 1

Is the manuscript scientifically sound in its present form?

Yes

Are the interpretations and conclusions justified by the results?

No

Is the language acceptable?

Yes

Do you have any ethical concerns with this paper?

No

Have you any concerns about statistical analyses in this paper?

No

Recommendation?

Accept with minor revision (please list in comments)

Comments to the Author(s)

This manuscript proposed a macroscopic engineering method to improve microwave absorption based on MXene material. This manuscript is inspired by previous graphene devices design, yet interesting for MXene EMA research. The following are some comments about this manuscript.

1. In figure 1, more material characterizations are required to support the claimed composite materials of $\text{TiO}_2\text{-Ti}_3\text{C}_2\text{T}_x$, and $\text{Fe}_3\text{O}_4@\text{TiO}_2\text{-Ti}_3\text{C}_2\text{T}_x$. For example, XPS mapping is necessary to be used to present the element distribution. The SEM image can not show the presence of Fe.
2. The references of MXene based functional devices need to be updated, see a recent review "X. Jiang, A.V. Kuklin, A. Baev, et al. Physics Reports 848 (2020) 1–58"
3. The experimental pictures of the MXene coated pyramid matrix are missing and should be presented in the manuscript. How is the quality of the coating? The performance of the devices highly depends on the fabrication process and outcome.
4. The authors claimed ultra-light MXene devices, which might be inaccurate and lack of experimental evidence. The $\text{Ti}_3\text{C}_2\text{T}_x$ multi-layer devices are not necessarily lighter than graphene-based devices.

Review form: Reviewer 2

Is the manuscript scientifically sound in its present form?

Yes

Are the interpretations and conclusions justified by the results?

Yes

Is the language acceptable?

Yes

Do you have any ethical concerns with this paper?

No

Have you any concerns about statistical analyses in this paper?

No

Recommendation?

Accept with minor revision (please list in comments)

Comments to the Author(s)

As a novel type of 2D materials, Mexene possesses important application prospect in field of microwave absorption materials. To enhance the microwave absorption width of Mexene-hybrid matrix materials, two types of Mexene hybrid materials were fabricated for the first time by macroscopic design approach through simulation method. The results are beneficial for the successful production of Mexene hybrid-matrix materials with wide microwave absorption bandwidth. I thus recommend its publication in this journal after minor revision as followings:

- 1) In the as-obtained $\text{Fe}_3\text{O}_4@\text{TiO}_2\text{-Ti}_3\text{C}_2\text{T}_x$, please provide the particular size of Fe_3O_4 nanoparticles, distribution, and proportion?
- 2) What is the interaction type between the Fe_3O_4 nanoparticles and Mexene matrix?

Decision letter (RSOS-200456.R0)

Dear Dr Tan:

Title: Enhancement of microwave absorption bandwidth of MXene nanocomposites through macroscopic design
Manuscript ID: RSOS-200456

Thank you for submitting the above manuscript to Royal Society Open Science. On behalf of the Editors and the Royal Society of Chemistry, I am pleased to inform you that your manuscript will be accepted for publication in Royal Society Open Science subject to minor revision in accordance with the referee suggestions. Please find the reviewers' comments at the end of this email.

The reviewers and handling editors have recommended publication, but also suggest some minor revisions to your manuscript. Therefore, I invite you to respond to the comments and revise your manuscript.

Because the schedule for publication is very tight, it is a condition of publication that you submit the revised version of your manuscript before 21-Jun-2020. Please note that the revision deadline will expire at 00.00am on this date. If you do not think you will be able to meet this date please let me know immediately.

Kind regards,
Dr Laura Smith
Publishing Editor, Journals

On behalf of the Subject Editor Professor Anthony Stace and the Associate Editor Dr Chaohua Cui.

RSC Associate Editor:
Comments to the Author:
(There are no comments.)

RSC Subject Editor:
Comments to the Author:
(There are no comments.)

Reviewer comments to Author:
Reviewer: 1

Comments to the Author(s)

This manuscript proposed a macroscopic engineering method to improve microwave absorption based on MXene material. This manuscript is inspired by previous graphene devices design, yet interesting for MXene EMA research. The following are some comments about this manuscript.

1. In figure 1, more material characterizations are required to support the claimed composite materials of $\text{TiO}_2\text{-Ti}_3\text{C}_2\text{Tx}$, and $\text{Fe}_3\text{O}_4@\text{TiO}_2\text{-Ti}_3\text{C}_2\text{Tx}$. For example, XPS mapping is necessary to be used to present the element distribution. The SEM image can not show the presence of Fe.
2. The references of MXene based functional devices need to be updated, see a recent review "X. Jiang, A.V. Kuklin, A. Baev, et al. Physics Reports 848 (2020) 1–58"
3. The experimental pictures of the MXene coated pyramid matrix are missing and should be presented in the manuscript. How is the quality of the coating? The performance of the devices highly depends on the fabrication process and outcome.

4. The authors claimed ultra-light MXene devices, which might be inaccurate and lack of experimental evidence. The Ti_3C_2Tx multi-layer devices are not necessarily lighter than graphene-based devices.

Reviewer: 2

Comments to the Author(s)

As a novel type of 2D materials, Mexene possesses important application prospect in field of microwave absorption materials. To enhance the microwave absorption width of Mexene-hybrid matrix materials, two types of Mexene hybrid materials were fabricated for the first time by macroscopic design approach through simulation method. The results are beneficial for the successful production of Mexene hybrid-matrix materials with wide microwave absorption bandwidth. I thus recommend its publication in this journal after minor revision as followings:

- 1) In the as-obtained $Fe_3O_4@TiO_2-Ti_3C_2Tx$, please provide the particular size of Fe_3O_4 nanoparticles, distribution, and proportion?
- 2) What is the interaction type between the Fe_3O_4 nanoparticles and Mexene matrix?

Author's Response to Decision Letter for (RSOS-200456.R0)

See Appendix A.

RSOS-200456.R1 (Revision)

Review form: Reviewer 1

Is the manuscript scientifically sound in its present form?

Yes

Are the interpretations and conclusions justified by the results?

Yes

Is the language acceptable?

Yes

Do you have any ethical concerns with this paper?

No

Have you any concerns about statistical analyses in this paper?

No

Recommendation?

Accept as is

Comments to the Author(s)

Acceptable now.

Decision letter (RSOS-200456.R1)

Dear Dr Tan:

Title: Enhancement of microwave absorption bandwidth of MXene nanocomposites through macroscopic design
Manuscript ID: RSOS-200456.R1

It is a pleasure to accept your manuscript in its current form for publication in Royal Society Open Science. The chemistry content of Royal Society Open Science is published in collaboration with the Royal Society of Chemistry.

On behalf of the Subject Editor Professor Anthony Stace and the Associate Editor Dr Chaohua Cui.

RSC Associate Editor:
Comments to the Author:
(There are no comments.)

RSC Subject Editor:
Comments to the Author:
(There are no comments.)

Reviewer(s)' Comments to Author:
Reviewer: 1

Comments to the Author(s)
Acceptable now.

Appendix A

Responses to the comments of the reviewers (RSOS-200456)

The authors thank the reviewers for their useful comments and suggestions. We have thoroughly addressed the comments of the reviewers and made suitable modifications in the revised manuscript. The comments of the reviewers are in italics and our responses are in normal font. The modifications to the revised manuscript are highlighted in yellow.

Reviewer: 1

Comments to the Author(s):

This manuscript proposed a macroscopic engineering method to improve microwave absorption based on MXene material. This manuscript is inspired by previous graphene devices design, yet interesting for MXene EMA research.

We thank the reviewer for appreciating this work and recommending its publication.

The following are some comments about this manuscript. In figure 1, more material characterizations are required to support the claimed composite materials of $\text{TiO}_2\text{-Ti}_3\text{C}_2\text{T}_x$, and $\text{Fe}_3\text{O}_4@\text{TiO}_2\text{-Ti}_3\text{C}_2\text{T}_x$. For example, XPS mapping is necessary to be used to present the element distribution. The SEM image can not show the presence of Fe.

Response: Both the nanocomposites viz., $\text{TiO}_2\text{-Ti}_3\text{C}_2\text{T}_x$, and $\text{Fe}_3\text{O}_4@\text{TiO}_2\text{-Ti}_3\text{C}_2\text{T}_x$ was synthesized according to the standard procedure adopted in the references [17] and [18], where all the characterizations were provided. In this work, the same synthetic procedure, the same conditions, and parameters of synthesis were used (the chemicals were also procured from the same suppliers). The objective of this work is more towards application point of view. However, as suggested by the reviewer, we added more elemental distribution information by providing EDX data in the revised manuscript (Supporting information). In addition, in the case of $\text{Fe}_3\text{O}_4@\text{TiO}_2\text{-Ti}_3\text{C}_2\text{T}_x$, the permeability data is predominant, which thus indicates the presence of Fe through magnetic Fe_3O_4 in $\text{TiO}_2\text{-Ti}_3\text{C}_2\text{T}_x$. Following modifications have been made in the revised manuscript,

The energy dispersive X-ray (EDX) confirms the presence of Fe (Figure S1, supporting information).

Figure S1. EDX spectra of $\text{Fe}_3\text{O}_4@\text{TiO}_2\text{-Ti}_3\text{C}_2\text{T}_x$ MXene.

2. The references of MXene based functional devices need to be updated, see a recent review "X. Jiang, A.V. Kuklin, A. Baev, et al. *Physics Reports* 848 (2020) 1–58"

Response: We thank the reviewer for this valuable suggestion. We have cited this paper in the revised manuscript. Following modifications have been made in the revised manuscript,

References:

[14] Jiang X, Kuklin A V., Baev A, Ge Y, Ågren H, Zhang H, et al. Two-dimensional MXenes: From morphological to optical, electric, and magnetic properties and applications. *Phys. Rep.* 2020; 848, 1-58.

#3. The experimental pictures of the MXene coated pyramid matrix are missing and should be presented in the manuscript. How is the quality of the coating? The performance of the devices highly depends on the fabrication process and outcome.

Reply: The fabrication of pyramidal and multi-layered pyramidal structure was carried out by a solution process and was schematically shown in the Fig.S1 and S2 (Supporting information). Since it was fabricated through an *in-situ* solution process, so the intrinsic electromagnetic parameters ought to be predominant in the material. However, there might be some minor structural errors in the fabrication. As suggested, we provided the experimental sample picture of multi-layered MXene coated pyramid matrix in the revised manuscript (Supporting information). We also agree with the reviewer and mentioned the point of structure fabrication error in the revised manuscript. Following modifications have been made in the revised manuscript:

The variation of experimental results as compared to the simulated results is believed to be due to the errors in raw data fitting and structure fabrication.

Figure S7. (a) Schematic of broadband microwave absorption measurement technique, (b) prepared multi-layer MXene hybrid-matrix pyramid design

PS: The experiment was carried out in the Microwave Laboratory, Department of Electrical and Communication Engineering (ECE), Indian Institute of Science (IISc), Bangalore, India. Please note that because of present Covid-19 situation, Microwave Lab is being closed till September, 2020. The picture shown in Figure S7. (b) was taken earlier and more photos are not available now due to the close of the laboratory. We believe that the reviewer has well understood the situation and will recommend this manuscript to publish with the proof of experiment picture.

4. The authors claimed ultra-light MXene devices, which might be inaccurate and lack of experimental evidence. The Ti₃C₂T_x multi-layer devices are not necessarily lighter than graphene-based devices.

Reply: We thank the reviewer for pointing out this typographical error in the manuscript. We have modified the sentence in the revised manuscript. Following modifications have been made in the revised manuscript,

MXene, the new family of 2D materials having numerous nanoscale layers, is being considered as a novel microwave absorption material.

Reviewer: 2

Comments to the Author(s):

As a novel type of 2D materials, Mexene possesses important application prospect in field of microwave absorption materials. To enhance the microwave absorption width of Mexene-hybrid matrix materials, two types of Mexene hybrid materials were fabricated for the first time by macroscopic design approach through simulation method. The results are beneficial for the successful production of Mexene hybrid-matrix materials with wide microwave absorption bandwidth. I thus recommend its publication in this journal after minor revision as followings:

We thank the reviewer for appreciating this work and recommending its publication.

1) In the as-obtained $Fe_3O_4@TiO_2-Ti_3C_2T_x$, please provide the particular size of Fe_3O_4 nanoparticles, distribution, and proportion?

Reply: We thank the reviewer for this valuable suggestion. The size of Fe_3O_4 nanoparticles was obtained to be ~ 8 nm. As suggested, histogram was provided in the supporting information. Following modification have been made in the revised manuscript.

The Fe_3O_4 nanoparticle size was recorded using ImageJ software from SEM image and histogram was shown in the **Figure S2 (Supporting information)**. The average particle size of the Fe_3O_4 nanoparticles was found to be ~ 8 nm.

Figure S2. Particle size distribution of Fe_3O_4 nanoparticles in the $TiO_2-Ti_3C_2T_x$ MXene.

2) *What is the interaction type between the Fe₃O₄ nanoparticles and MXene matrix?*

Reply: The interaction between Fe₃O₄ nanoparticles and MXene matrix as synthesized composite was explained elaborately in the references [17] and [18]. Since the objective of this work is towards application point of view, not synthetic oriented, we did not explain it previously. As suggested, the following modification have been made in the revised manuscript:

The interaction of Fe₃O₄ nanoparticles with MXene was explained in Ref. [17]. The exposed hydroxyl groups offer the possibility of binding with MXenes along with blending metal-oxygen stretching modes such as Fe–O and Ti–O [17].